# TopoFit: Rapid Reconstruction of Topologically-Correct Cortical Surfaces

**Andrew Hoopes**[1]                                   AHOOPES@MGH.HARVARD.EDU

**Juan Eugenio Iglesias**[1,2,3,4]                    JIGLESIASGONZALEZ@MGH.HARVARD.EDU

**Bruce Fischl**[1,2,3,5]                              BFISCHL@MGH.HARVARD.EDU

**Douglas Greve**[*1,2]                                DGREVE@MGH.HARVARD.EDU

**Adrian V. Dalca**[*1,2,3]                            ADALCA@MIT.EDU

[1] *Martinos Center for Biomedical Imaging, Massachusetts General Hospital*

[2] *Department of Radiology, Harvard Medical School*

[3] *Computer Science and Artificial Intelligence Lab, Massachusetts Institute of Technology*

[4] *Centre for Medical Image Computing, University College London*

[5] *Harvard-MIT Division of Health, Sciences, and Technology*

**Editors:** Under Review for MIDL 2022

## Abstract

Mesh-based reconstruction of the cerebral cortex is a fundamental component in brain image analysis. Classical, iterative pipelines for cortical modeling are robust but often time-consuming, mostly due to expensive procedures that involve topology correction and spherical mapping. Recent attempts to address reconstruction with machine learning methods have accelerated some components in these pipelines, but these methods still require slow processing steps to enforce topological constraints that comply with known anatomical structure. In this work, we introduce a novel learning-based strategy, TopoFit, which rapidly fits a topologically-correct surface to the white-matter tissue boundary. We design a joint network, employing image and graph convolutions and an efficient symmetric distance loss, to learn to predict accurate deformations that map a template mesh to subject-specific anatomy. This technique encompasses the work of current mesh correction, fine-tuning, and inflation processes and, as a result, offers a $150\times$ faster solution to cortical surface reconstruction compared to traditional approaches. We demonstrate that TopoFit is $1.8\times$ more accurate than the current state-of-the-art deep-learning strategy, and it is robust to common failure modes, such as white-matter tissue hypointensities.

**Keywords:** Cortical Surface Reconstruction, Topology, Geometric Deep-Learning

## 1. Introduction

Many medical image analysis techniques employ mesh representations, as opposed to image-based segmentation maps, for the reconstruction (or fitting) of tissue boundaries around regions of interest. This mesh-based approach is especially relevant in brain MRI processing, where polygonal meshes are often leveraged for reconstructing and studying the cerebral cortex, the outer layer of brain tissue. This layer is characterized by the topology of a 2D sheet, and due to its highly-folded nature, image-space Cartesian distances between points on the cortex can substantially underestimate the true distances along the sheet. Establishing a mesh-based surface representation of the cortex is therefore a fundamental

---

* Contributed equally

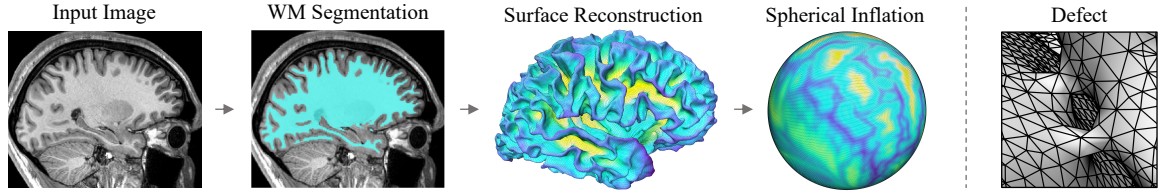

Figure 1: **Left:** Partial overview of the traditional surface-based cortical processing pipeline, involving white-matter (WM) segmentation, mesh reconstruction, and spherical inflation. **Right:** Example of a topological defect in which anatomically implausible "bridges" are incorrectly estimated between two gyri.

step for downstream and multi-modal analysis of its structure, connectivity, function, and disease pathology.

Classical approaches to cortical reconstruction are robust and well established in neuroimaging pipelines, but tend to involve long runtimes: tens of minutes or even hours to run on just a single image (Fischl, 2012; Gaser and Dahnke, 2016). A substantial portion of this runtime is spent fulfilling the requirement to represent the cortex with a topologically-correct, genus zero manifold, facilitating anatomical plausibility and downstream geometric computations. However, surfaces are generally derived by tessellating volumetric segmentations, providing no guarantee that this condition is met. To address topological errors, cortical modeling tools employ mesh correction strategies that consume a substantial and variable amount of time (Fischl et al., 2001; Ségonne et al., 2007; Yotter et al., 2011).

In recent years, rapid deep-learning solutions have dominated the field of brain MRI analysis for a variety of tasks, but only a small handful of methods tackle cortical surface placement (Cruz et al., 2021; Ma et al., 2021). While these methods offer runtime improvements for specific components of surface-fitting pipelines, they are not complete solutions and require substantial pre- or post-processing to remove topological errors. Thus, cortical surface reconstruction remains a major step in brain MRI analysis without an established rapid, learning-based solution.

In this work, we address this missing link by introducing TopoFit, a novel approach that rapidly fits an accurate and topologically-correct polygonal mesh to the cerebral cortex. By leveraging convolutions in both the image and graph domains, TopoFit learns to estimate deformations from a mesh template, yielding a detailed reconstruction of subject-specific white-matter surfaces in less than 20 seconds on a CPU. The graph network facilitates deformations informed by the geometry and topology of the manifold. We demonstrate considerable robustness to local deviations in image intensity, which often disrupt traditional reconstruction methods, and we establish a 45.8% improvement in accuracy compared to a recent deep-learning approach. Our code is available at https://github.com/ahoopes/topofit, and we also publicly distribute our method within the open-source FreeSurfer package (Fischl, 2012). While TopoFit was developed in the context of neuroimaging, the concepts introduced here can be applied in a wide range of medical imaging contexts.

## 2. Related Work

**Classical Surface Analysis.** Surface-based neuroimaging software, such as FreeSurfer (Fischl, 2012) and CAT12 (Gaser and Dahnke, 2016), construct detailed, per-hemisphere models of cortical gray-matter to facilitate analysis of functional activation, diffusion connectivity, and structural brain morphometry. As illustrated in Figure 1, the traditional surface-modeling pipeline begins by segmenting interior white-matter tissue from a brain MRI image then tessellating the computed segmentation map. After correcting for topological defects, this mesh is fine-tuned using a per-vertex optimization that considers the underlying image intensity gradient while constraining properties of the mesh geometry and topology (Dale et al., 1999). The resulting white-matter surface is subsequently deformed outwards to model the exterior pial tissue boundary. While these steps yield a complete representation of cortical gray-matter at the subject-level, a universal mapping must be computed for further group-level analysis and surface segmentation. This correspondence is achieved in a spherical coordinate system by inflating the white-matter manifold to the geometry of a sphere (Fischl et al., 1999a), which enables a curvature-matching registration with an average atlas (Fischl et al., 1999b; Yeo et al., 2009) and, consequently, establishes a global anatomical mapping across surfaces.

**Topology Correction.** A mesh can only be diffeomorphically inflated and spherically-mapped if it possesses a topology that is homotopic, or continuously deformable, to that of a sphere. Specifically, a topologically-correct surface can be defined as a convex polyhedron with an Euler characteristic of two (genus zero). Since the initial white-matter surface derives from a segmentation with discrete resolution and potential for error, topological defects in the surface connectivity are common. These errors manifest as anatomically-implausible holes in the white-matter (Figure 1) and bridges connecting two banks of a sulcus. Defects must be corrected using time-consuming "manifold-surgery" algorithms that take into account the underlying geometry and image information to appropriately *cut* or *fill* regions of the mesh (Fischl et al., 2001; Ségonne et al., 2007; Yotter et al., 2011). In FreeSurfer, topology correction consumes roughly 70% of the total white-matter surface reconstruction time.

Recent research directions explore topology in deep-learning by leveraging differential persistent homology (PH), a technique to compute topological features across a landscape of spatial resolutions (Edelsbrunner et al., 2000). PH has been used to regularize model weights (Gabrielsson et al., 2020) as well as classification decision boundaries (Chen et al., 2019), and it is particularly useful for enforcing topological priors in medical image segmentations (Byrne et al., 2020; Clough et al., 2019). While TopoFit and PH methods both involve constraining the topology of model outputs, PH aims to encourage topological correctness during training, whereas our method avoids the need to correct the topology all together by using a topologically-correct, predefined mesh.

**Learning-Based Reconstruction.** Recent deep-learning efforts have concentrated on mesh segmentation and registration (Hao et al., 2020; Lyu et al., 2021; Wu et al., 2019; Zhao et al., 2021) of the cortex, but few focus on the mesh placement itself. The leading method, DeepCSR (Cruz et al., 2021), uses implicit surface representations (Park et al., 2019) to reconstruct both white-matter and pial surfaces, but it requires a time-consuming topology correction. PialNN (Ma et al., 2021) uses explicit surfaces to project the pial boundary,

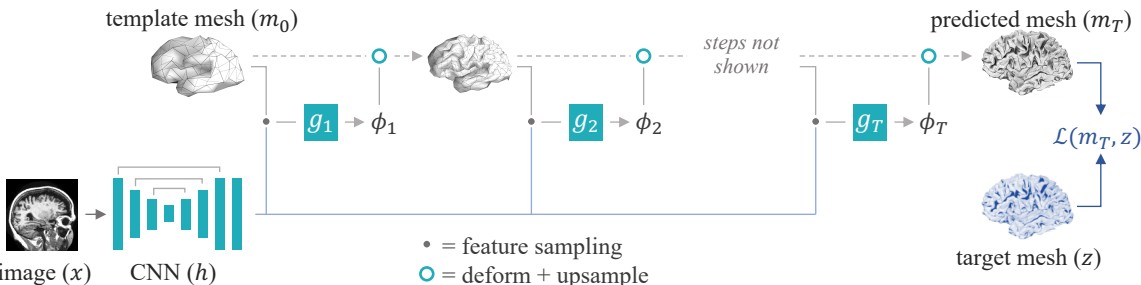

Figure 2: TopoFit architecture: a series of graph-convolutional (GCN) blocks iteratively warp a template surface $m_0$ to fit the anatomy of an input image $x$. Each block $g_t$ predicts a deformation $\phi_t$ by sampling spatial image features from a volumetric U-Net $h$ at the positions of $m_{t-1}$. The template mesh is up-sampled between each deformation step.

but it still relies on a pre-computed, corrected white-matter mesh. Similar to our method, Voxel2Mesh (Wickramasinghe et al., 2020) addresses general anatomical segmentation by deforming a spherical template to the relevant tissue boundary, guaranteeing topologically-correct outputs, but it is limited to the focus of broad structure segmentation. Our work also builds on concepts from learning-based image (Balakrishnan et al., 2019; de Vos et al., 2019; Mok and Chung, 2020) and point-cloud registration (Hansen et al., 2019; Shen et al., 2021; Wang and Solomon, 2019) tactics that employ networks to align data pairs.

## 3. Method

Given a 3D brain MR image $x$, we aim to estimate a high-resolution, topologically-correct 2D manifold $y$ of the white-matter. We employ a function $f_\theta(x, m) = \phi$ that computes a deformation field $\phi$ from a template surface $m$ to the subject-specific anatomy in $x$. By using a template $m$ with predefined, correct connectivity, the deformed mesh reconstruction is guaranteed to be free of intrinsic topological defects. We design $f_\theta(\cdot, \cdot)$ as a convolutional network that employs both image-space and graph convolutions, at several scales. We find the optimal network parameters using

$$\hat{\theta} = \arg\min_\theta \Big[ \, \mathbb{E}_{(x,z)\sim\mathcal{D}} \, \big[ \, \mathcal{L}_{dist}(y, z) + \lambda \mathcal{L}_{reg}(y) \, \big] \, \Big], \tag{1}$$

where $y = m \circ f_\theta(x, m)$, with $\circ$ representing the deformation procedure, and $z$ is the target surface for an image $x$ in the training set $\mathcal{D}$. The loss term $\mathcal{L}_{dist}$ measures surface similarity, and $\mathcal{L}_{reg}$, weighted by the hyperparameter $\lambda$, encourages geometric regularity of $y$.

### 3.1. Architecture

We design $f_\theta$ as a network that jointly leverages convolutional operations on the image grid and on the graph of the template mesh (Figure 2). Specifically, a first network $h_{\theta_h}(x)$, with input image $x$, employs image-space convolutions to extract useful features on the image

grid. A second network $g_{\theta_g}(m; h_{\theta_h}(x))$ employs blocks of graph convolutions that operate on the features identified by $h_{\theta_h}$, interpolated at intermediate locations of the template mesh $m$, and output a series of mesh deformations $\phi$.

To preserve geometric regularity, we adopt a scale-space strategy for the GCN $g_{\theta_g}$. We employ a low-resolution template mesh $m_0 = m$ derived from a high-resolution icosphere mesh, corresponding to the average geometry across a set of surfaces. At each deformation step $t = \{1..T\}$, a GCN *block* $g_t$ takes as input the $h_{\theta_h}$ feature activations sampled at the spatial coordinates of $m_{t-1}$, and predicts the corresponding deformation $\phi_t$. We then form $m_t = u(m_{t-1} \circ \phi_t)$, where $u(\cdot)$ is a mesh upsampling operation based on the connectivity of the icosphere, such that $y = m_T$. We illustrate this framework in Figure 2.

### 3.2. Loss Function

We minimize a symmetric surface distance between the estimated mesh $y$ and the ground truth $z$. The symmetric Chamfer distance (Fan et al., 2017) is frequently used in existing methods, but is often computationally impractical for two high-resolution manifolds (with more than $150k$ points each). Instead, we take advantage of topological consistency across all training surfaces, which are aligned to a spherical cortical template and ensure that a given vertex represents the same anatomical region for all surfaces. This enables us to precompute a neighborhood $\mathcal{N}_v$ of $k$ closest vertices for each vertex $v \in \mathcal{V}$ and efficiently minimize the localized symmetric distance between the predicted and target surface:

$$\mathcal{L}_{dist}(y, z) = \frac{1}{2|\mathcal{V}|} \sum_{v \in \mathcal{V}} \min_{n \in N_v} ||p_v^y - p_n^z|| + \min_{n \in N_v} ||p_v^z - p_n^y||, \qquad (2)$$

where $p_i^j$ is the Cartesian coordinate of $i$-th vertex of mesh $j$.

We also employ geometric regularization that minimizes self-intersecting faces to encourage an anatomically-accurate, smooth manifold. Specifically, we use a *hinge-spring* term that maximizes the angle between all neighboring triangles $a, b$ in the set of mesh edges $E$:

$$\mathcal{L}_{reg}(y) = \frac{1}{|E|} \sum_{a,b \in E} (1 - (\hat{u}_a \cdot \hat{u}_b))^2, \qquad (3)$$

where $\hat{u}_i$ is the unit normal of the $i$-th triangle in $y$. This metric encourages smoothness without shrinking the mesh towards the center of mass, a side-effect present when using a vertex-based spring term (Fischl et al., 1999a).

### 3.3. Implementation

We implement a U-Net-like convolutional network $h$ (Ronneberger et al., 2015), comprised of four down-sampling and up-sampling convolutional layers with skip connections, followed by one more convolutional layer. Each layer involves 64 channels and LeakyReLU activations.

In our experiments we use $T = 7$ GCN blocks $g_t$, each with a similar U-Net-like architecture employing edge-convolutions (Wang et al., 2019). Given the input activations of $h$ sampled at vertices $m_t$, each block $g_t$ comprises at most three down-sampling and up-sampling LeakyReLU-activated layers, each with 64 channels, and a final, linearly activated layer with three channels that estimates $\phi_t$. Resolution levels of the template surface

are defined by the orders of icosphere tessellation, which facilitates up- and down-sampling of graph features using a universal adjacency map. In our experiments, the template $m_0$ resolution corresponds to icosphere order 1 (42 vertices), and the final target mesh $m_T$ corresponds to order 7 (163,842 vertices). In this configuration, the $g_t$ U-Nets at $t = \{1, 2\}$ cannot down-sample beyond icosphere order 1 and therefore contain only one and two resolution levels, respectively. We use vertex neighborhood size $k = 100$, which maximizes surface placement accuracy as determined by a grid hyperparameter search.

During training, we sample an image and corresponding target surface $(x, z)$ from $\mathcal{D}$ at each mini-batch and augment $x$ with random added Gaussian noise (max $\sigma = 0.1$). We use the Adam optimizer (Kingma and Ba, 2014) with an initial learning rate of $10^{-4}$, which is reduced by a factor of two for every 10,000 iterations that do not exhibit a decrease in validation loss. We use the TensorFlow Graphics (Abadi et al., 2016; Valentin et al., 2019), Keras (Chollet et al., 2015), and Neurite (Dalca et al., 2018) packages, conducting all experiments using Intel Xeon Silver 4214R CPUs and Nvidia RTX 8000 GPUs.

## 4. Experiments

We evaluate the ability of TopoFit to accurately and rapidly reconstruct white-matter surfaces. We compare to the recent state-of-the-art deep-learning method and test robustness to common sources of error for conventional vertex placement techniques. Lastly, we perform an analysis of model hyperparameters and predicted mesh regularity.

**Data:** We employ a set of 1,232 subjects with T1-weighted MRI scans gathered from the publicly-available OASIS (Marcus et al., 2007), IXI[1], MCIC (Gollub et al., 2013), and Buckner40 (Fischl et al., 2002) cohorts. We split these data into train, validate, and test subsets of sizes 788, 40, and 404, respectively, and additionally hold out the Buckner40 subjects entirely until final evaluation. For each scan, we use FreeSurfer v7.2 to derive a bias-corrected input image, an affine alignment to the Talairach atlas space, and white-matter surfaces for both hemispheres of the brain. For all training surfaces, we remesh to the universal icosphere topology, mapped by their computed spherical alignments with the FreeSurfer cortical atlas. We conform per-hemisphere images to $1mm$ isotropic voxel sizes with intensities normalized between $[0, 1]$ and crop them to a $96 \times 144 \times 208$ region.

**Evaluation:** We train two models, one for each hemisphere, and evaluate their performance for the entire set of test subjects, using manually-corrected FreeSurfer surfaces as a "silver-standard" reference set. We further compare TopoFit to DeepCSR, the primary established learning-based approach to white-matter reconstruction. We train and evaluate DeepCSR following online instructions and source code, adapting the training scheme to account for our data augmentation protocol and learning-rate decay schedule to yield the best DeepCSR results.

**Metrics:** We measure reconstruction accuracy using the mean, 99th-percentile (P99), and max (Hausdorff) symmetric distances between predicted and reference surfaces. Since surface modeling near the medial wall is noisy and unused in cortical analysis, we only include vertices for cortical regions marked by FreeSurfer. We also report the Dice overlap (Dice, 1945) between the interior of the filled surfaces, using 0.75 $mm$ voxel sizes. Finally, we quantify surface regularity by the percent of self-intersecting faces in the mesh.

---

1. Acquired from http://brain-development.org/ixi-dataset.

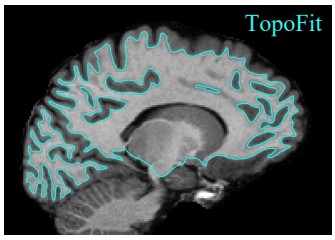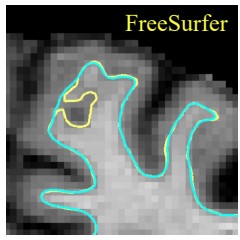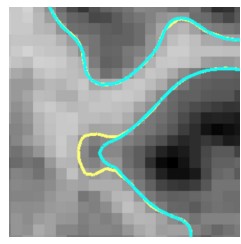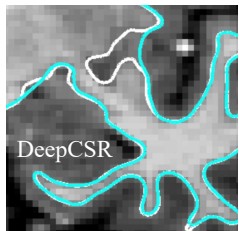

Figure 3: Representative reconstruction examples for TopoFit (blue), FreeSurfer (yellow), and DeepCSR (white). FreeSurfer and DeepCSR tend to get stuck in local minima around unexpected white-matter tissue abnormalities.

## 4.1. Reconstruction Accuracy

In the first experiment, we evaluate the ability of our proposed method to fit a topologically-correct surface to the white-matter boundary. Table 1 shows that TopoFit computes highly accurate surfaces and substantially outperforms DeepCSR, which tends to collapse into or broadly extend the white-matter in the presence of tissue abnormalities or intensity variation (Figure 3). The performance of TopoFit is consistent across each subject set (Table A1), generalizing well to the *unseen* Buckner40 cohort and on MCIC subjects. To fit white-matter surfaces to both hemispheres of the brain, TopoFit requires $17.3 \pm 0.2$ total seconds on a CPU and $1.2 \pm 0.1$ seconds on a GPU. This is in stark contrast to FreeSurfer, which requires $1,086.6 \pm 852.0$ CPU seconds for the same process. Furthermore, since our method leverages the pre-defined topology of an icosphere, it can avoid the additional $1,518.3 \pm 750.6$ seconds required for spherical mapping, resulting in a roughly $150 \times$ total speed-up over FreeSurfer. We find similar runtime improvements compared to DeepCSR, as shown in Table 1.

## 4.2. Robustness

A disadvantage of iterative mesh optimization techniques is their tendency to get stuck in local minima near regions of unexpected intensity distribution, requiring end-users to manually inspect each surface and correct errors. We test whether TopoFit is robust to these failure modes by evaluating on 20 randomly-selected test surfaces that required manual edits during the preprocessing stage. In this experiment, we determine accuracy by using a human rater to indicate whether our method makes the same mistake as FreeSurfer and by measuring the mean symmetric distance to the fixed reference surface within a 4 $mm$ radius of the error region. We find that TopoFit produces correct white-matter reconstructions

| | Mean Dist. ($mm$) | P99 Dist. ($mm$) | Hausdorff ($mm$) | Dice (%) | Runtime ($s$) |
|---|---|---|---|---|---|
| DeepCSR | $0.24 \pm 0.09$ | $1.56 \pm 1.13$ | $7.53 \pm 2.01$ | $96.0 \pm 1.2$ | $818.2 \pm 134.9$ |
| **TopoFit** | $\mathbf{0.13 \pm 0.07}$ | $\mathbf{0.78 \pm 0.35}$ | $\mathbf{4.35 \pm 1.41}$ | $\mathbf{97.6 \pm 0.6}$ | $\mathbf{1.2 \pm 0.1}$ |

Table 1: White-matter surface placement accuracy and GPU runtime for both hemispheres.

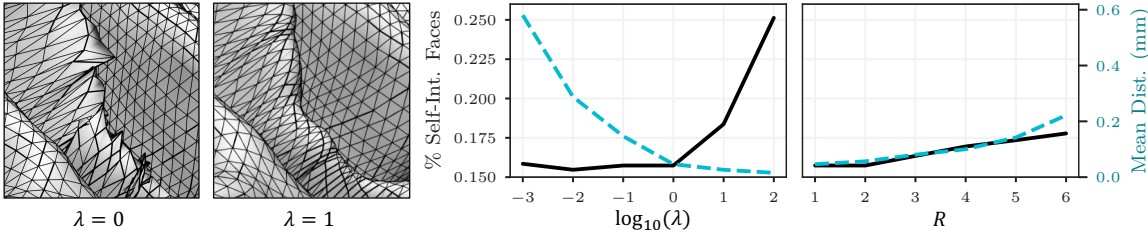

Figure 4: Model results given different hyperparameter values. The optimal regularization weight $\lambda$ and template resolution level $R$ aim to maximize surface placement accuracy while constraining the number of self-intersecting faces in the mesh.

in 90% (18/20) of these regions, resulting in a negligible $0.18 \pm 0.04$ $mm$ local deviation from the corrected reference. As highlighted in Figure 3, we find that FreeSurfer often fails to correctly fit surfaces in regions where white-matter lesions neighbor the cortex, whereas TopoFit is able to account for the global spatial context of the image.

### 4.3. Mesh Regularity and Hyperparameters

In addition to surface placement accuracy, maximizing mesh regularity is important for reproducing realistically smooth manifolds and ensuring that mesh connectivity corresponds to underlying anatomy. We perform a hyperparameter grid search to determine the optimal $\mathcal{L}_{reg}$ smoothing weight $\lambda^*$. Figure 4 shows that the optimal $\lambda^* = 1.0$, used in our models, enforces a smooth mesh and minimizes the amount of self-intersecting faces (only 0.04%) without sacrificing reconstruction accuracy. Using $\lambda = 0$ results in predicted surfaces in which 5.1% of faces are self-intersecting. We also investigate how scale-space architecture choices impact mesh regularity by training models that use a different icosphere order $R$ for the template mesh at the first GCN step $m_0$. We find that a scale-space approach helps predict regular and accurate meshes and hypothesize that the decrease in performance as $R$ increases is the result of far-reaching initial deformations that tangle nearby vertices.

### 5. Conclusion

We introduce TopoFit, a learning-based solution for reconstructing cerebral white-matter folds with a topologically-correct manifold. In a matter of seconds on a CPU, TopoFit can accomplish the tasks of traditional reconstruction pipelines that often require 40 minutes or more, by deforming a template mesh with intrinsic mapping to a sphere. This template surface is fit to the subject with high accuracy, even in regions that commonly lead to errors in FreeSurfer. We plan to extend our framework to include a pial surface reconstruction, and by further exploring the integration of learning-based cortical segmentation and spherical registration methods with TopoFit, we also plan to create an end-to-end model that encompasses the entirety of the cortical processing workflow. Lastly, we aim to extend this strategy to reconstruct the tissue boundaries of other, subcortical brain structures.

## Acknowledgments

Support for this research was provided by the National Institute for Biomedical Imaging and Bioengineering (P41 EB015896, 1R01 EB023281, R01 EB006758, R21 EB018907, R01 EB019956, P41 EB030006), the National Institute on Aging (1R56 AG064027, 1R01 AG064027, 5R01 AG008122, R01 AG016495, 1R01 AG070988), the BRAIN Initiative Cell Census Network (U01 MH117023), the National Institute of Mental Health (R01 MH123195, R01 MH121885, 1RF1 MH123195), the National Institute for Neurological Disorders and Stroke (R01 NS0525851, R21 NS072652, R01 NS070963, R01 NS083534, 5U01 NS086625, 5U24 NS10059103, R01 NS105820), the NIH Blueprint for Neuroscience Research (5U01 MH093765), and Shared Instrumentation Grants 1S10 RR023401, 1S10 RR019307, and 1S10 RR023043.

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

## Appendix A.

**Buckner40 (40 test subjects)**

|  | Mean Dist. ($mm$) | P99 Dist. ($mm$) | Hausdorff ($mm$) | Dice (%) |
|---|---|---|---|---|
| DeepCSR | $0.25 \pm 0.17$ | $1.60 \pm 1.27$ | $7.64 \pm 2.12$ | $95.95 \pm 2.04$ |
| TopoFit | $0.11 \pm 0.02$ | $0.73 \pm 0.37$ | $4.34 \pm 1.45$ | $97.69 \pm 0.51$ |

**IXI (158 test subjects)**

|  | Mean Dist. ($mm$) | P99 Dist. ($mm$) | Hausdorff ($mm$) | Dice (%) |
|---|---|---|---|---|
| DeepCSR | $0.26 \pm 0.10$ | $1.37 \pm 0.74$ | $7.47 \pm 1.80$ | $95.90 \pm 1.38$ |
| TopoFit | $0.15 \pm 0.09$ | $0.81 \pm 0.37$ | $4.41 \pm 1.55$ | $97.37 \pm 1.25$ |

**OASIS (130 test subjects)**

|  | Mean Dist. ($mm$) | P99 Dist. ($mm$) | Hausdorff ($mm$) | Dice (%) |
|---|---|---|---|---|
| DeepCSR | $0.22 \pm 0.07$ | $1.71 \pm 1.43$ | $7.60 \pm 2.21$ | $96.29 \pm 0.87$ |
| TopoFit | $0.11 \pm 0.02$ | $0.68 \pm 0.27$ | $4.31 \pm 1.36$ | $97.81 \pm 0.41$ |

**MCIC (76 test subjects)**

|  | Mean Dist. ($mm$) | P99 Dist. ($mm$) | Hausdorff ($mm$) | Dice (%) |
|---|---|---|---|---|
| DeepCSR | $0.25 \pm 0.06$ | $1.55 \pm 0.75$ | $7.45 \pm 1.82$ | $95.76 \pm 0.86$ |
| TopoFit | $0.15 \pm 0.05$ | $0.92 \pm 0.35$ | $4.25 \pm 0.98$ | $97.30 \pm 0.73$ |

Table A1: Dataset-specific comparison of total white-matter reconstruction accuracy for both hemispheres of the brain.

