# OpenReview forum: "TopoFit: Rapid Reconstruction of Topologically-Correct Cortical Surfaces"
_MIDL.io/2022/Conference — MIDL 2022_

### Official Review · Reviewer_WpSm · 2022-01-23

**Confidence:** 5
**Preliminary Rating:** 4
**Recommendation:** Poster

**Summary:**

The authors are addressing an old well known problem that is still not so well addressed in mesh processing. In constructing (brain) mesh as 2D polygons, the resulting mesh often introduces topological artifacts. They come up with a loss based method and compared their method against deep learning.

**Strengths:**

The claim their method improves 45.8% over deep learning based method (DeepCSR). Also they distributed code through github. They plan to incorporate into FreeSurfer package. The method is applied to large-scale brain surface mesh data.

**Weaknesses:**

The authors are not utilizing the state-of-arts in topological data analysis (persistent homology) that were applied to various medical imaging problems. Even they call their loss function as topological, their loss functions are still all geometric. Thus, they are comparing geometric loss against another geometric method (deep learning).

**Deanonymize Review:**

no

**Detailed Comments:**

For topological loss based on persistent homology, see

@inproceedings{chen2019topological,
  title={A topological regularizer for classifiers via persistent homology},
  author={Chen, Chao and Ni, Xiuyan and Bai, Qinxun and Wang, Yusu},
  booktitle={The 22nd International Conference on Artificial Intelligence and Statistics},
  pages={2573--2582},
  year={2019},
  organization={PMLR}
}

@inproceedings{songdechakraiwut2021topological,
  title={Topological learning and its application to multimodal brain network integration},
  author={Songdechakraiwut, Tananun and Shen, Li and Chung, Moo},
  booktitle={International Conference on Medical Image Computing and Computer-Assisted Intervention},
  pages={166--176},
  year={2021},
  organization={Springer}
}

@article{clough2019topological,
  title={A topological loss function for deep-learning based image segmentation using persistent homology},
  author={Clough, James R and Byrne, Nicholas and Oksuz, Ilkay and Zimmer, Veronika A and Schnabel, Julia A and King, Andrew P},
  journal={arXiv preprint arXiv:1910.01877},
  year={2019}
}

Page 2. Authors said the method can be applied to a wide range of medical imaging problems. Need to be more specific and tell readers where we can possibly apply the proposed method beyond the problem setting (mesh topology correction).

**Paper Type:**

both

**Questions To Address In The Rebuttal:**

Their loss function is still geometric and comparing their method against geometric deep learning method.
They should compare it to state-of-arts in persistent homological methods. Also they are comparing their method to only one method. There must be many different methods for mesh topology corrections developed over the years.



**Special Issue:**

no

---

### Official Review · Reviewer_i1Ba · 2022-01-24

**Confidence:** 4
**Preliminary Rating:** 5
**Recommendation:** Oral

**Summary:**

This paper present an algorithm to fit meshes to white matter surface on the cortex, with state of the art accuracy, less error sensitive to reduce manual correction work and faster computational speed.
The main strength of the proposed algorithm is to avoid the use of expansive post-processign step to correct the mesh topology to ensure it does not contain holes. In addition, the resulting mesh contains less errors usually manually corrected.

**Strengths:**

The paper is clearly written, with a detailed and extensive literature review in the introduction. The method does not have to many details on the standard machine learning technique, but the general concepts are cleanly explained. The examples and results show well the strengths of this approach and possible future works. The method seems novel and of good practical use in an applied context due to its accuracy and computational speed.

**Weaknesses:**

I did not find any major weaknesses in this work. My only main comment would be to present in more detail the differences with the two other methods (DeepCSR and FreeSurfer) which makes the presented method better. It is clear that the topology correction step is the crucial point, but it was not very clear to me what in the proposed algorithm avoided the use of this step. A more detailed explanation (even if technical) would help the exposition of the paper.

**Deanonymize Review:**

no

**Detailed Comments:**

I don't have minor comments.

**Final Rating After The Rebuttal:**

5: Strong Accept

**Justification Of The Final Rating:**

All good for me! .......................................... Sorry for the dots, I didn't know what to write to get to 200 chars...........................................................................

**Paper Type:**

both

**Questions To Address In The Rebuttal:**

I think it is a good paper to be accepted as is, so I don't have many questions here.
As stated above, one possible improvement would be to add more details on how the presented architecture ensures that the resulting mesh has no holes, as compared with previous algorithms which do not have this feature.

**Special Issue:**

yes

---

### Meta-Review · Area_Chair_KsFB · 2022-02-15

**Recommendation:** Accept (Oral)
**Confidence:** 5

**Metareview:**

This paper presents a deep learning algorithm for fitting a topologically correct white matter surface by deforming a template surface to fit to the individual.

Reviewers agreed that the paper was well written, with some comments that some aspects of the methods and the motivations could be clearer. In general, all agree that is is a sound solution to a relevant problem that warrants inclusion at MIDL.

One thing that I think is missing is a clearer description of how the method improves relative to Ma 2021, which is also achieves topologically correct surfaces through a deformable model.

---

### Decision · Program_Chairs · 2022-02-28

Accept